# MALDI-TOF MS for malaria vector surveillance: A cost-comparison analysis using a decision-tree approach

Jonathan Karisa[1,2]*, Cassidy Rist[3], Mercy Tuwei[1], Kelly Ominde[1], Brian Bartilol[1], Zedekiah Ondieki[1], Haron Musani[1], Caroline Wanjiku[1], Joseph Mwangangi[1], Charles Mbogo[1], Martin Rono[1], Philip Bejon[1,4], Marta Maia[1,4]

1 Kenya Medical Research Institute, Wellcome Trust Research Programme, Kilifi, Kenya, 2 The Open University, United Kingdom, 3 Virginia-Maryland College of Veterinary Medicine at Virginia Tech 205 Duck Pond Drive Blacksburg, Virginia, United States of America, 4 University of Oxford, Centre for Global Health and Tropical Medicine, Oxford, United Kingdom

* karisajonna@gmail.com

## Abstract

### Background

The use of MALDI-TOF MS for mosquito identification and surveillance is routinely used in developed countries as an affordable alternative to molecular methods. However, in low- and middle-income countries (LMIC) where mosquito-borne diseases carry the greatest burden, the method is not commonly employed. Using the Kenyan national malaria program (NMCP) as a case study, we compared the costs of current methods used for malaria vector surveillance to those that would be incurred if MALDI-TOF MS were used instead.

### Methods

A deterministic decision tree analytic model was developed to systematically calculate the costs associated with materials and labour, and time-to-results for two workflows, i.e., current molecular methods versus MALDI-TOF MS. The analysis assumed an annual sample size of 15,000 mosquitoes (representing the average number of mosquitoes analysed annually by the Kenyan NMCP) processed at a local laboratory in Kenya.

### Findings

We estimate that if the Kenyan national entomological surveillance program shifted sample processing completely to MALDI-TOF MS, it would result in 74.48% net time saving, up to 84% on material costs and 77% on labour costs, resulting in an overall direct cost savings of 83%.

**Data availability statement:** All the data is provided as Supporting information.

**Funding:** MM-Investment 057212-Bill and Melinda Gates Foundation. The funder did not play any role in the study design, data collection and analysis, decision to publish, or preparation of the manuscript.

**Competing interests:** The authors have declared that no competing interests exist.

## Interpretation

Adoption of MALDI-TOF MS for malaria vector surveillance can result in substantial time and cost savings. The ease of performance, the rapid turn-around time, and the modest cost per sample may bring a paradigm shift in routine entomological surveillance in Africa.

## Introduction

Entomological surveillance is a critical component of mosquito-borne disease control programs. It involves the systematic and continuous collection, identification, and analysis of mosquitoes to track changes in vector populations [1]. Currently, in Kenya and many other sub-Saharan malaria endemic countries, the National Malaria Control Programs (NMCPs) conduct regular surveys in sentinel sites and rely on partner laboratories to process the collected mosquitoes. The current workflow of procedures involves initial morphological identification of mosquitoes to species complex followed by transportation to a laboratory for processing or storage. In the laboratory, molecular methods including polymerase chain reaction (PCR) and enzyme-linked immunosorbent assay (ELISA) are used to identify the species [2,3], presence of sporozoite [4–6], and blood meal source [7,8]. Less frequently, whilst still in the field, microscopy is used for age-grading through parity dissections [9,10]. Limited technical capacity and funding result in only small proportion of mosquitoes being analysed. Consequently, valuable opportunities to gain insights into local vector dynamics are missed. Addressing this limitation could enable programs to have better understanding of malaria transmission dynamics, evaluate how vector populations are responding to current interventions or environmental changes, detect invasive species, and therewith generate real-time information to support decision-making.

MALDI-TOF MS (matrix-assisted laser desorption ionization time-of-flight mass spectrometry) is a powerful analytical tool for protein profiling that has been extensively applied for rapid and cost-effective speciation of microbiological specimens [11]. Its fast turn-around time, technical ease, accuracy and low-cost per sample have revolutionised microbiology [12]. In European countries, this tool has been used for mosquito surveillance for over a decade [13–15] and is part of the European Centre for Disease Prevention and Control (ECDC) guidelines for the surveillance of invasive mosquitoes [16]. However, in sub-Saharan Africa where the burden of mosquito borne disease is highest its application is lagging despite the equipment being readily available in many African microbiology laboratories and National Public Health Institutes. Using a cost-effective and robust tool like MALDI-TOF MS would significantly benefit malaria-endemic countries, where limited resources constrain the generation of high-quality surveillance data. A cost-comparison analysis evaluating the workflow of current approaches versus MALDI-TOF MS for entomological surveillance in a malaria endemic country may help establish a solid evidence base supporting its recommendation for mosquito surveillance in SSA.

## Materials and methods

### Study design

A cost comparison analysis of the conventional methods currently used by the NMCP for entomological malaria surveillance in Kenya (i.e., current workflow) to the use of MALDI-TOF MS (i.e., proposed workflow) was performed using a deterministic decision tree analytic model. The costs associated with materials (reagents and consumables), labour costs, and total direct costs (material and labour cost combined), as well as time-to-results were compared for both workflows measuring three key entomological parameters (species identification, Plasmodium detection and blood meal analysis). Results were based on an analysis of 15,000 mosquito samples corresponding to the approximate number of samples analysed by the Kenyan NMCP in one year and the experience of using MALDI-TOF MS for entomological research at the Kenya Medical Research Institute (KEMRI) in Kilifi. We considered that laboratories had already purchased the equipment used in both workflows for research purposes; therefore, the initial outlay/start-up costs associated with the different workflows were not included in this analysis. Moreover, the maintenance cost was also excluded from the analysis as we assumed the partner laboratories would be responsible for the cost of maintaining their own equipment, shared by multiple programs across disciplines.

### Current workflow

In the current workflow, individual mosquitoes are dissected into three different compartments: legs and wings, head and thorax, and abdomen (for blood-fed mosquitoes). The legs and wings are subjected to DNA extraction for species identification by PCR [2,3]. If samples are designated as unidentifiable, some may be further subjected to bi-directional Sanger sequencing to determine species [17]. The cephalothorax compartment is used for *Plasmodium* detection by circumsporozoite (CSP-) ELISA [18]. All positive samples are confirmed with a second CSP-ELISA whereby the remaining ELISA lysate/homogenate is heated to 100° C for 10 minutes to eliminate false positivity associated with zoophilic mosquito species [18–20]. In the case of blood-fed mosquitoes, the abdomens are used for analysis of blood meal source from humans and common household livestock by ELISA [8]. Any undetected samples are not usually analysed further because of the unavailability of conjugated antibodies for the numerous host possibilities.

### Proposed workflow using MALDI TOF MS

For the proposed workflow, individual mosquitoes are dissected into three different compartments: legs and wings, head and thorax, and abdomen (for blood-fed mosquitoes). The head and thorax are transversely divided into two halves; the first half for spectral acquisition by MALDI-TOF MS is used to simultaneously measure three entomological parameters viz., species identification, *Plasmodium* detection and parity status determination [21]. Given current gap in MALDI-TOF MS reference libraries for malaria infection detection, the second half of the head and thorax is reserved for *Plasmodium* sporozoite detection by ELISA when needed for quality assurance purposes [18]. Similarly, the legs and wings for those samples are subjected to DNA extraction and screened for species identification using PCR [2,3], followed by the use of bi-directional Sanger sequencing [17] for quality assurance purposes. Individual abdomens of blood-fed mosquitoes are crushed in 50 µl of LC-MS water: 10 µl is used for MALDI-TOF MS blood meal source analysis, and the remainder is held for ELISA when needed for quality assurance purposes.

### Material costs

A micro-costing approach was used to determine the material costs for the current and proposed workflows (S1–S6 Table). All pricing corresponds to a 2022–2023 price list. For each workflow, we estimated the reagent and consumable costs (*i.e.*, material costs) by identifying all the resources used in each assay and their respective unit costs. We derived unit costs from supplier invoices and KEMRI Wellcome Trust Research Programme (KWTRP) procurement databases.

For PCR, ELISA, and sequencing assays, we determined the cost per sample per assay by dividing the total materials cost per assay by 90, the number of mosquito samples typically loaded in one plate. We assumed all the primers had 20 nucleotide bases, and the average volume of water to be added to make a concentration of 100pmol/µl was 500 µl. For MALDI-TOF MS, we determined the cost per sample by dividing the total materials cost by 47 mosquito samples, the number of mosquito samples typically loaded in one plate. We excluded additional reagent costs associated with MALDI-TOF MS re-analysis because they were negligible, as the technologist only needed to pick the sample spot to be re-analysed, and the instrument automatically acquired the spectra from the sample. We determined the total material costs for analysing 15,000 mosquitoes in each workflow by applying the material cost per sample to each step in the decision tree analytic model.

### Time-to-results

Time-to-results is the time taken from sample preparation to obtaining all results in each workflow, which includes time to perform primary tests, supplementary tests for samples designated as unidentified or undetected, and data interpretation (S7 Table). To estimate this, we first estimated a time-to-results per sample per assay. For each workflow, two independent staff members were observed and timed as they completed each component of the workflow. For the current workflow, staff analysed 96 samples (90 samples and six controls) per plate; therefore, the time-to-results per sample per assay was calculated by averaging the time per assay and then dividing the average time by 90 samples. For the proposed workflow, staff analysed 47 mosquito samples and two controls (positive – BTS and negative – matrix only) per plate; therefore, the average time-to-results per sample per assay was calculated by averaging the time per assay and dividing the average time by 47 samples. The time also included what was required for MALDI target plate cleaning. The total time-to-results for analysing 15,000 samples in each workflow was determined by applying the time-to-results per sample to each step in the decision tree analytic model (Fig 1).

### Labour costs

Labour costs per assay were calculated using the estimate of the time-to-results per sample per assay with the following adjustment: exclusion of time for incubation periods above 60 minutes, as it is assumed that staff members can be engaged in other activities unrelated to the assay being conducted. Labour costs per sample per assay were calculated using the mean base lab tech salary of 6,289.82 United Stated Dollars (USD) annually (0.55 USD per minute), based on a 40-hour work week, and multiplied by the adjusted time-to-results per sample per assay. Total labour costs for analysing 15,000 samples in each workflow were determined by applying the labour cost per sample to each step in the decision tree analytic model. The salary was based on the KEMRI salary scale in the Kenyan Salaries and Remuneration Commission (SRC) approved structure.

### Direct costs

The total direct cost per sample per assay was determined as follows:

$$\text{Direct cost} = \text{Material cost per sample per assay} + \text{Labor cost per sample per assay}$$

Total direct costs for analysing 15,000 samples in each workflow were determined by applying the labour cost per sample to each step in the decision tree analytic model.

### Decision tree analytic model

A decision tree analytic model for cost comparison (Fig 1) was constructed using Amua software (version 0.3.0; https://github.com/zward/Amua). For each workflow, we assumed that the number of blood-fed mosquitoes was approximately

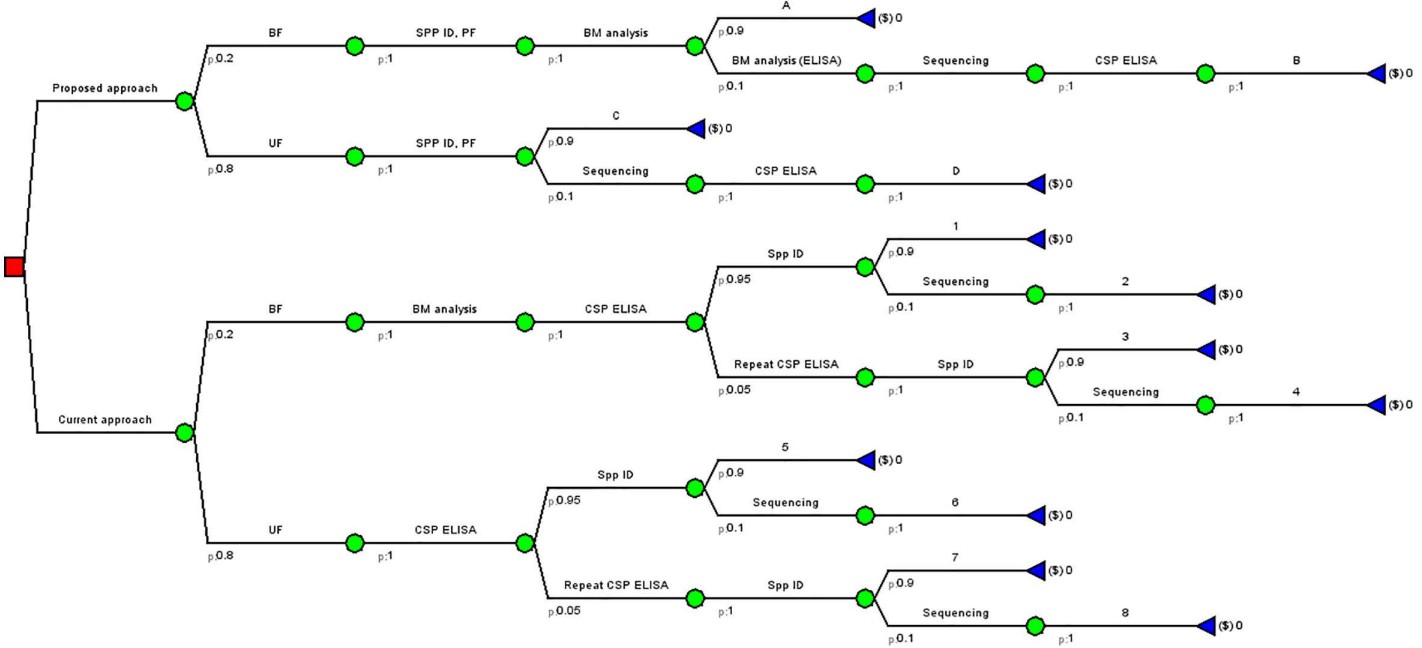

**Fig 1. Decision tree analytic model for the proposed MALDI-TOF MS workflow and current workflow in entomological surveillance as performed by the Kenyan NMCP (Abbreviations: BF – Blood fed; UF – Unfed; SPP ID – Species identification; PF – Plasmodium falciparum detection, PS – Parity status; CSP ELISA – Circumsporozoite Protein Enzyme-linked immunosorbent assay; MALDI – Matrix-assisted laser desorption ionization–time-of-flight mass spectrometry; BM – Blood meal analysis).**

20% of the total collection. We also estimated that 10% of the samples processed for species identification and blood meal sources by both workflows might give unsatisfactory results, prompting additional or other methods to categorise the mosquitoes. This 10% estimate for PCR and ELISA is consistent with what is seen in the laboratory, and for MALDI-TOF MS reflects the current deficiencies in the reference database and sample quality impacted by poor storage conditions [21]. In the current workflow, we assumed that approximately 5% of all the samples screened for *Plasmodium* would be positive and would undergo a second CSP ELISA to confirm the positivity status by heating the lysates before the assay.

A deterministic cohort simulation of 15,000 samples was run, considering its various probability nodes. All costs were converted to USD, with the conversion from Kenyan Shilling and Great Britain pounds (GBP) to USD using a currency converter: (https://www.oanda.com/currency-converter/en/), accessed on May 16, 2023.

The decision tree model was run four times:

1. Applying time-to-results per sample per assay to obtain total time to analyse 15,000 samples per workflow.

2. Applying materials cost per sample per assay to obtain total materials cost for analysis of 15,000 mosquito samples.

3. Applying labour cost per sample per assay to obtain total labour cost for analysis of 15,000 mosquito samples.

4. Applying direct cost per sample per assay to obtain total direct cost for analysis of 15,000 mosquito samples.

Moreover, to perform a supplementary analysis, we constructed an additional decision tree analytic model (Fig 2), assuming an ideal scenario where all the entomological endpoints are acquired using MALDI-TOF MS with 100% reportable results. This additional 'ideal scenario' model was developed as we foresee the potential for 100% reportable results given the development of a comprehensive reference database and improvements in storage conditions that will eliminate poor-quality spectra or flatlines.

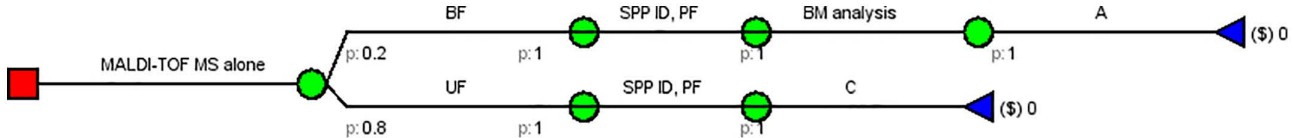

**Fig 2. Decision tree analytic model for the MALDI-TOF MS 'ideal scenario' workflow, where 100% of results are reportable (Abbreviations: BF – Blood fed; UF – Unfed; SPP ID – Species identification; PF – Plasmodium falciparum detection, MALDI – Matrix-assisted laser desorption ionization–time-of-flight mass spectrometry; BM – Blood meal analysis).**

## Results

### Total time-to-results

**Time per sample per assay.** The estimated time required to analyse one sample in the current workflow for each assay was 2.620 minutes (min) for species identification by PCR, 15.958 min for follow-up species identification by Sanger sequencing, 4.245 min for *Plasmodium* detection by CSP-ELISA, and 3.900 min for blood meal source determination by ELISA (Table 1). Analysing one mosquito sample by MALDI-TOF MS required 2.010 min, which is inclusive of obtaining results for three entomological parameters, viz., species identification, parity status and *Plasmodium* infection status detection. For blood meal identification by MALDI-TOF MS, it required the same time (2.010 min) to obtain results, including the time needed for target plate cleaning. The adjusted time-to-analysis per sample per assay, which is used in the labour cost analysis, is also provided in Table 1. Times are adjusted to exclude incubation periods above 60 minutes, as it is assumed that staff members can be engaged in other activities unrelated to the assay being conducted.

### Final comparison of total time

Considering the present analysis plan of the Kenyan NMCP where the assays performed include species identification, Plasmodium sporozoite detection and blood meal source identification, total time-to-results for the 15,000 samples in the current workflow was estimated at 141,795.75 minutes (2,363.2625 hrs. or ~ 99 days) compared to 67,654.50 min (1,127.575 hrs or ~ 47 days) in the proposed workflow which reflects an 52.29% net time savings of 74,141.25 min (1,235.69 hrs. or ~ 52 days) (Table 2). This means that, on average, the total time-to-results per sample is 9.4531 min for the current workflow, compared to 4.5103 min for the proposed workflow.

### Cost comparison analysis

**Materials cost per sample per assay.** We estimated the materials cost per sample analysis for PCR species identification to be 1.49 USD from DNA extraction to gel electrophoresis. The materials cost for re-analysing a single

**Table 1. Summary of the material, labour cost per sample and time taken to analyse one sample for each of the assay. The adjusted time excludes incubation periods greater than 60 minutes.**

| Procedure | Time per sample (min) | Adjusted time (min) | Materials cost per sample (USD) | Labour cost per sample (USD) |
|---|---|---|---|---|
| MALDI-TOF MS (blood meal) | 2.01 | 2.01 | 0.59 | 0.11 |
| PCR (species ID) | 2.62 | 0.62 | 1.49 | 0.3 |
| Sanger sequencing (follow-on species ID) | 15.958 | 12.624 | 23.23 | 0.51 |
| CSP ELISA (Plasmodium) | 4.245 | 1.579 | 0.3 | 0.12 |
| Repeated CSP ELISA, heated lysates (follow-on Plasmodium) | 4.245 | 4.245 | 0.3 | 0.12 |
| ELISA (blood meal) | 3.9 | 1.234 | 1.21 | 0.06 |

**Table 2. Summary of time-to-results for the proposed MALDI-TOF MS workflow and current workflow as performed by the Kenyan NMCP.**

|  | Time (min) per sample | Time (min) for 15000 samples |
|---|---|---|
| Current | 9.4531 | 141795.75 |
| Proposed | 4.5103 | 67654.50 |
| Time saving | 4.9428 | 74141.25 |

sample by Sanger sequencing was approximated at 23.23 USD. The materials cost for CSP ELISA for *Plasmodium* detection and blood meal ELISA for blood meal source discrimination was 0.30 USD and 1.21 USD, respectively (Table 1). A detailed breakdown of the costs of reagents and consumables for standard testing is shown in S1–S6 Tables. The materials cost per sample analysis by MALDI-TOF MS was approximated at 0.59 USD, which covers sample preparation (0.25 USD) and plate cleaning (0.35 USD). This did not differ between the analysis performed on the head and thorax or the abdomen. A detailed breakdown of the costs of reagents and consumables for MALDI-TOF MS is shown in S6 Table.

**Final comparison of materials cost.** Considering the present analysis plan of the Kenyan NMCP, the current workflow would cost an estimated amount of 65,550.00 USD compared to 46,300.00 USD in the proposed workflow for the analysis of 15,000 samples, reflecting a cost savings of 19,250 USD (Table 3). This means that, on average, the total materials cost per sample is 4.37 USD for the current workflow, compared to 3.09 USD for the MALDI-TOF MS workflow.

**Labour cost per sample per assay.** The adjusted time-to-analysis per assay (Table 1) was used to calculate labour cost per sample per assay. The labour costs per sample per assay were estimated to be 0.30, 0.12, 0.06, and 0.51 USD for PCR species identification, CSP ELISA, ELISA (blood meal) and sequencing, respectively (Table 1; Appendix 7). For MALDI-TOF MS, labour cost per sample per assay was estimated to be 0.11 USD per sample for each respective assay, i.e., head and thorax analysis (species identification, infection status determination and parity status determination) and blood meal analysis (Table 1).

**Final comparison of labour cost.** Considering the present analysis plan of the Kenyan NMCP, the current workflow would cost an estimated amount of 7,335.00 USD compared to 2,952.00 USD in the proposed workflow for the analysis of 15,000 samples, reflecting a cost savings of 4,383 USD (Table 3). This means that, on average, the total labour cost per sample is 0.48 USD for the current workflow, compared to 0.20 USD for the proposed workflow.

**Final cost comparison of the two workflows (direct cost).** Considering the present analysis plan of the Kenyan NMCP, the current workflow would cost an estimated amount of 72,885.00 USD compared to 49,252.00 USD in the proposed workflow for the analysis of 15,000 samples, reflecting a cost savings of 23,633.00 USD (Table 3).

**Investigation of other scenario (MALDI-TOF MS ideal scenario).** Under the "ideal" scenario, it takes 36,180.0 min (603 hrs or 26 days) to analyse 15,000 mosquito samples, which reflects a 74.48% net time savings of 105615.75 min (1760.26 hrs. or 74 days) as compared to the current workflow (Table 4). Moreover, analysing 15,000 mosquito samples under the ideal scenario would cost an estimated 10,620.00 USD in materials and 1,980.00 USD in labour costs, which equates to an overall direct cost savings of 62,025.00 USD or 83.12% as compared to the current workflow (Table 4).

**Table 3. Summary of the materials costs, labour costs, and total direct costs between the current and proposed workflows for entomological surveillance for the proposed scenario in the current Kenyan NMCP.**

|  | One sample (USD) | | 15000 samples (USD) | | Overall cost saving (USD) | % saving |
|---|---|---|---|---|---|---|
|  | Current | Proposed | Current | Proposed | Cost saving |  |
| Materials | 4.37 | 3.09 | 65,550.00 | 46,300.00 | 19,250.00 | 29.39 |
| Labour | 0.48 | 0.20 | 7,335.00 | 2,952.00 | 4,383.00 | 59.75 |
| Direct | 4.86 | 3.29 | 72,885.00 | 49,252.00 | 23,633.00 | 32.43 |

**Table 4. The material, labour, total direct costs, and time-to-results between the current and ideal scenario workflows for entomological surveillance.**

| | 15000 samples | | Overall savings | |
| --- | --- | --- | --- | --- |
| | *Current* | *Proposed* | *Savings* | *% saving* |
| Materials cost (USD) | 65,730.0 | 10,620.0 | 55,110.0 | 83.84 |
| Labour cost (USD) | 8,895.0 | 1,980.0 | 6,915.0 | 77.74 |
| Direct cost (USD) | 74,625.0 | 12,600.0 | 62,025.0 | 83.12 |
| Time (min) | 141,795.75 | 36,180.00 | 105,615.75 | 74.48 |

## Discussion

Here we report the results of a cost comparison analysis of the current approach for entomological surveillance that primarily relies on PCR and ELISA versus our proposed system that primarily uses MALDI-TOF MS in a case study of the Kenyan NMCP. To the best of our knowledge, this is the first study comparing the cost of laboratory procedures supporting malaria vector surveillance using MALDI TOF MS and standard molecular methods. Given that entomological surveillance provides multiple endpoints, a cost analysis that accounts for all steps in the diagnostic process offers a complete picture of actual costs incurred and allows for a better comparison between current and novel approaches. This information helps decision-makers as they face difficult choices when considering adding new technologies to their programs.

Shifting from molecular methods to MALDI-TOF MS would reduce the time-to-results, material costs, labour costs, and direct costs per sample. Using decision tree analysis, we demonstrate that when analysing the 15,000 mosquitoes collected annually by NMCP, they would save approximately 19,430.00 USD (29.56%) on materials and 5,943.00 USD (66.81%) on labour costs, resulting in an overall direct cost savings of 25,373.00 USD (34.0%). These results demonstrate the cost and time efficiency of MALDI-TOF MS and indicate that if programs adopt this strategy they would save money and decision-makers would have faster access to the data.

Even though MALDI-TOF may provide cost-savings, technical teams need to be confident about the reliability of the results. MALDI-TOF MS has above 97.5% accuracy in discriminating sibling species of *An. gambiae* and *An. funestus* s.l. [21,22]. Moreover, MALDI-TOF MS overcomes the challenge of misclassification in the field as the unknown sample is queried against a comprehensive spectral database of different mosquito species and not just a selection of complex siblings [23,24].

In the case of *Plasmodium* detection, MALDI-TOF MS has been shown to have acceptable sensitivity (92.8%) and specificity (100%) using insectary-reared and artificially-infected *An. stephensi* [25], whereas ELISA has a relatively low sensitivity (78.9%) when compared to microscopy [18], and a high false positivity rate, which has been reported predominantly in zoophilic vectors [19]. The collection of reference spectra from field caught sporozoite-infected mosquitoes is ongoing. We expect within the next year to have created databases for *Plasmodium falciparum* detection in wild-caught mosquitoes. MALDI-TOF device does not come with a built-in mosquito vector database and that such a database must be constructed separately or adopted as the databases are available online for adoption.

MALDI-TOF MS has the potential to revolutionize the field of medical entomology, as it has in medical microbiology. We anticipate that, with the development of a comprehensive MALDI-TOF MS reference database for majority if not all the vectors in Africa, including infection and parity status and blood meal sources, this approach could streamline entomological analyses and reduce reliance on supplementary tests.

## Limitations of the study

The major limitation to this study is that we used a deterministic decision tree model, which does not allow for variation in materials, labour, or time costs, which could affect our estimate measures. Although there is probably a range of cost

savings we would anticipate, given the significant cost savings identified in the proposed workflow, a stochastic model, which allows for variation in cost and time estimates, would likely still produce favourable cost savings results.

## Conclusion

MALDI-TOF MS represents an innovative technology for rapid and accurate simultaneous characterisation of mosquito species, *Plasmodium* infection status, parity status, and blood meal sources. Moreover, it provides a platform that significantly reduces costs for the laboratory's sample processing, materials, and labour. Despite the high capital cost of the instrument, the ease of performance, the rapid turn-around time to results, and the modest cost of testing for each sample make this new methodology a paradigm shift for entomological surveillance. As work continues in upgrading and validating additional mosquito species (secondary malaria, exotic and arboviral vectors) in the MALDI-TOF MS databases and methods of sample storage improve, we anticipate improved results at a significantly lower cost.

## Supporting information

**S1 Table. Cost analysis of reagents and consumables used in species identification by polymerase chain reaction.**
(PDF)

**S2 Table. Cost analysis of reagents and consumables used in species identification by Sanger sequencing.**
(PDF)

**S3 Table. Cost analysis of reagents and consumables used in infection status determination by enzyme linked immunosorbent assay.**
(PDF)

**S4 Table. Cost analysis of reagents and consumables used in blood meal analysis by enzyme linked immunosorbent assay (ELISA).**
(PDF)

**S5 Table. Cost analysis of reagents and consumables used in parity status determination by dissection.**
(PDF)

**S6 Table. Cost analysis of reagents and consumables used in the MALDI-TOF MS assay.**
(PDF)

**S7 Table. Time and cost analysis per sample for a laboratory technician.**
(XLSX)

## Acknowledgments

We would like to thank members of the Kenyan NMCP for their time and availability to provide information on the activities conducted. This paper has been published with the permission of the Director of the Kenya Medical Research Institute (KEMRI).

## Author contributions

**Conceptualization:** Jonathan Karisa, Cassidy Rist, Martin Rono, Philip Bejon, Marta Maia.

**Data curation:** Jonathan Karisa, Cassidy Rist.

**Formal analysis:** Jonathan Karisa, Cassidy Rist.

**Funding acquisition:** Cassidy Rist, Joseph Mwangangi, Martin Rono, Marta Maia.

**Investigation:** Jonathan Karisa, Cassidy Rist, Mercy Tuwei, Kelly Ominde, Brian Bartilol, Zedekiah Ondieki, Haron Musani, Caroline Wanjiku, Joseph Mwangangi, Philip Bejon, Marta Maia.

**Methodology:** Jonathan Karisa, Cassidy Rist, Mercy Tuwei, Kelly Ominde, Brian Bartilol, Zedekiah Ondieki, Haron Musani, Caroline Wanjiku, Joseph Mwangangi, Martin Rono, Philip Bejon, Marta Maia.

**Project administration:** Caroline Wanjiku, Joseph Mwangangi, Martin Rono, Marta Maia.

**Supervision:** Cassidy Rist, Caroline Wanjiku, Joseph Mwangangi, Charles Mbogo, Martin Rono, Marta Maia.

**Writing – original draft:** Jonathan Karisa.

**Writing – review & editing:** Cassidy Rist, Mercy Tuwei, Kelly Ominde, Brian Bartilol, Zedekiah Ondieki, Haron Musani, Caroline Wanjiku, Joseph Mwangangi, Charles Mbogo, Martin Rono, Philip Bejon, Marta Maia.

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
