## [Decision Letter · Decision Letter 0]

3 Oct 2025

Dear Dr. Karisa,

Thank you for submitting your manuscript to PLOS ONE. After careful consideration, we feel that it has merit but does not fully meet PLOS ONE’s publication criteria as it currently stands. Therefore, we invite you to submit a revised version of the manuscript that addresses the points raised during the review process.

We look forward to receiving your revised manuscript.

Kind regards,

Bersissa Kumsa, DVM, MSc, PhD

Academic Editor

PLOS ONE

Journal Requirements:

Additional Editor Comments:

Dear Authors,

The reviewers have completed their evaluation of your manuscript. I encourage you to revise and resubmit your work, ensuring that all reviewer comments are thoroughly addressed. Please incorporate the feedback carefully and provide a detailed, point-by-point response that clearly outlines every change made in response to the reviewers’ suggestions.

In addition, kindly correct all typographical and grammatical errors, and ensure that the manuscript is prepared in full compliance with the journal’s formatting and submission guidelines.

We look forward to receiving your revised submission.

Reviewers' comments:

Reviewer's Responses to Questions

**Comments to the Author**

1. Is the manuscript technically sound, and do the data support the conclusions?

Reviewer #1: Yes

Reviewer #2: Yes

2. Has the statistical analysis been performed appropriately and rigorously?

Reviewer #1: Yes

Reviewer #2: Yes

3. Have the authors made all data underlying the findings in their manuscript fully available?

Reviewer #1: Yes

Reviewer #2: No

4. Is the manuscript presented in an intelligible fashion and written in standard English?

Reviewer #1: Yes

Reviewer #2: Yes

Reviewer #1: The study is original and provides a comprehensive analysis of the costs of several techniques used in the monitoring and control of malaria. The MALDI-TOF technique is relatively new in the fields of entomology and parasitology, and the mosquito databases are developed in specialized laboratories rather than supplied by the companies that market the device. The technique is very advantageous in terms of cost, but time is needed to create a feasible database. With minor revisions, the article can be published.

Reviewer #2: In general, the manuscript is well written, concise, and informative while results and figures reflect the work. As described by the authors, the study brings valuable information into the cost associated with MALDI-TOF which could directly impact research performed in areas lacking access to this technology.

However, some overstatements and missing references might lead to speculative messages which must be avoided. All are described below for edits prior to publication.

Tables 1 and 3 contain same results for MALDI-TOF blood meal and species ID. Why not combining both conditions?

Lines 34-35 – Please edit to “... modest cost per sample may bring a paradigm shift in routine entomological surveillance in Africa.”

Line 40 – Remove: “..., and regular...”

Line 42 – Add “the” to “... the National Malaria...”

Line 76 – Which 4 entomological parameters? Please add those to methods, not only to the results portion of the manuscript.

Line 76 – Assumed? It was 15,000 samples, that is not subjective. Please delete “assumed”.

Line 78 – Your “own experience”? Please avoid subjective terminology.

Lines 79-82 – Assumed? Your study has considered that every laboratory had already the instrument, not assumed they had it. Please rephrase.

Line 111 – Add to the main text the timeline related to the price estimates used in the study. It could be listed as: all pricing corresponds to a 2025 price list... This information can be easily referenced in the future by the scientific community.

Line 130 – I understand the use of 2 staff members to perform the work. That wouldn’t make the cut for a true significant average, ideally you would have a minimum of 3 people performing the tasks. Please discuss that. Also, add a brief description of the level of expertise/training of each individual had prior to the study. If you had a highly experienced and a less experienced staff member performing the work, it should he mentioned.

Line 145 – Why the labor cost considered the US salary and not the SSA base salary? It would be more appropriate for the application at the SSA, correct? Any additional cost comparison between them?

Line 157-161 – Do you have a reference to sustain the 10% undetermined results? Avoid speculation. If that is in-house data, add that to the results and discuss it.

Lines 179-180 – Random figure 1 legend. It was confusing to see the incomplete legend in the middle of the text.

Line 229 – BM? Please edit table 2 on “blood meal” for consistency.

Line 245 – Anything else besides Table 2?

Line 259 – Investigation of a single hypothetical scenario, correct? Please remove the plural from the title.

Line 239 – Remove bold from Direct on Table 4.

Lines 262 and 265 - Table 5 typo.

Line 266 – Correct to: “...the material, labor, total direct costs, and time-to-results...”. Also remove the bold from Table 5.

Line 287 – Is it sensitivity or accuracy? Please reword.

Line 299 – Bold statement. Due to several limitations regarding processing of complex samples (3-component samples – pathogen-vector-host), the use of MALDI-TOF for any vector-borne has several complicators not part of most medical microbiology systems. Please rephrase accordingly.

**Do you want your identity to be public for this peer review?** For information about this choice, including consent withdrawal, please see our Privacy Policy

Reviewer #1: No

Reviewer #2: No

---

## [Author Response · Author response to Decision Letter 1]

8 Oct 2025

Article Review

PLOS ONE

MALDI-TOF MS for malaria vector surveillance: a cost-comparison analysis using a decision-tree approach

The article evaluates the use of MALDI-TOF MS as an alternative to molecular methods (PCR, ELISA, sequencing) for malaria vector surveillance in Kenya. The authors employ a decision-tree analytical model, with an annual volume of 15,000 processed mosquitoes, to compare material costs, personnel costs, and turnaround time.

The results show that by adopting MALDI-TOF MS, the national surveillance program could achieve:74.48% time savings, up to 84% savings in material costs, 77% savings in labor costs, a total of 83% direct cost savings. The authors conclude that MALDI-TOF MS may bring about a paradigm shift in entomological surveillance in Africa.

The strengths of the study lie in its high practical relevance – the article addresses a central issue in malaria control: the sustainability of vector surveillance in resource-limited countries.

The described methodology is clear, employing a deterministic decision-tree model that is well justified and easy to follow.

Costs are compared in a very detailed manner – the analysis includes materials, labor, direct costs, and time, providing a comprehensive picture.

The data are applicable to public policies – the results can be used by decision-makers to optimize health budgets.

An additional important strength is the study’s global relevance – although centered on Kenya, the conclusions can be extrapolated to other malaria-endemic countries.

Limitations and critical observations

Incomplete MALDI-TOF database – the method still has limitations in detecting Plasmodium infections and requires confirmation by ELISA, which reduces immediate practical advantages. Moreover, the article does not specify which database was used for mosquito detection, nor whether the database is already complete or still requires validation and expansion.

A near-complete MALDI-TOF database exists for the An. gambiae (including An. gambiae s.s., An. arabiensis, An. merus, and An. quadriannulatus) and An. funestus (including An. funestus s.s., An. rivulorum, and An. leesoni) complexes — the major malaria vectors found in Kenya. The database used in this study is described in detail in reference [21].

Exclusion of maintenance and investment costs – the analysis ignores initial investments (equipment, staff training), which may underestimate the real costs of implementation in resource-limited countries. The MALDI-TOF device is expensive, and the amortization period compared to a classic PCR line should be specified.

We agree with the reviewer that the MALDI-TOF device represents a significant capital investment. However, in our study, we did not include the initial investment or maintenance costs. This is because the device is already in use within microbiology laboratories for microbial identification, and its acquisition and upkeep have been budgeted for accordingly. Our approach was to repurpose existing MALDI-TOF infrastructure for entomological surveillance, thereby avoiding additional investment or operational costs specific to this application.

Limited field validation – MALDI-TOF sensitivity and specificity for Plasmodium detection are based on controlled experiments rather than large field datasets.

Thank you for this valuable comment. We agree that much of the validation for Plasmodium detection has relied on laboratory-infected mosquitoes, primarily because naturally infected field specimens are relatively rare. To address this limitation, we are now collaborating with several research groups to obtain a sufficient number of field-collected, Plasmodium-infected mosquitoes for MALDI-TOF MS database development and further validation.

Generalizability – the study relies on the Kenyan NMCP context; its exact applicability in other countries may vary depending on existing infrastructure.

We agree with your observation. Infrastructure availability indeed varies between countries. However, where MALDI-TOF MS devices are already in place—typically for clinical or microbiological applications—these can be repurposed for entomological surveillance with minimal additional investment. Furthermore, the establishment of regional reference laboratories could help mitigate infrastructure limitations by providing centralized support and access to MALDI-TOF MS technology for surrounding areas.

In conclusion The article is well-written, rigorous, and makes a significant contribution to the field of entomological surveillance and cost optimization in malaria control. The results have practical value and are supported by a clear methodology.

Recommendation

Publication is recommended after minor revisions, in order to clarify development plans for the MALDI-TOF database for Plasmodium and for all mosquito species. It should also be specified that the MALDI-TOF device does not come with a built-in mosquito vector database and that such a database must be constructed separately for each region

This has now been clearly detailed in the discussion

Reviewer #2:

In general, the manuscript is well written, concise, and informative while results and figures reflect the work. As described by the authors, the study brings valuable information into the cost associated with MALDI-TOF which could directly impact research performed in areas lacking access to this technology.However, some overstatements and missing references might lead to speculative messages which must be avoided. All are described below for edits prior to publication.

Tables 1 and 3 contain same results for MALDI-TOF blood meal and species ID. Why not combining both conditions?

Table 1 and 3 have been combined into one table; Table 1

Lines 34-35 – Please edit to “... modest cost per sample may bring a paradigm shift in routine entomological surveillance in Africa.”

Lines 34-35 in the abstract section has been revised accordingly.

Line 40 – Remove: “..., and regular...”

Line 40 has been revised accordingly

Line 42 – Add “the” to “... the National Malaria...”

Line 42 has been revised accordingly

Line 76 – Which 4 entomological parameters? Please add those to methods, not only to the results portion of the manuscript.

Line 76 has been revised to include the parameters measured. It now reads three key entomological parameters (species identification, Plasmodium detection and blood meal analysis)

Line 76 – Assumed? It was 15,000 samples, that is not subjective. Please delete “assumed”.

Line 76 has been revised accordingly. The term assumed has been deleted.

Line 78 – Your “own experience”? Please avoid subjective terminology.

Line 78 has been revised accordingly.

Lines 79-82 – Assumed? Your study has considered that every laboratory had already the instrument, not assumed they had it. Please rephrase.

Lines 79-82 has been revised by replacing the word assumed with considered.

Line 111 – Add to the main text the timeline related to the price estimates used in the study. It could be listed as: all pricing corresponds to a 2025 price list... This information can be easily referenced in the future by the scientific community.

In Line 111 has been revised accordingly, this statement has been added: All pricing corresponds to a 2022 - 2023 price list.

Line 130 – I understand the use of 2 staff members to perform the work. That wouldn’t make the cut for a true significant average, ideally you would have a minimum of 3 people performing the tasks. Please discuss that. Also, add a brief description of the level of expertise/training of each individual had prior to the study. If you had a highly experienced and a less experienced staff member performing the work, it should he mentioned.

Thank you for your comment. We agree that having a minimum of three individuals would provide a more statistically robust estimate of the time required to perform the assays. However, due to logistical constraints, we were limited by the number of available staff. Only two individuals were available to carry out this exercise: a PhD student with extensive experience in the procedure and a laboratory technician with moderate experience. We believe their combined expertise provides a reasonable representation of the expected time required under typical laboratory conditions.

Line 145 – Why the labor cost considered the US salary and not the SSA base salary? It would be more appropriate for the application at the SSA, correct? Any additional cost comparison between them?

The labor cost was based on the KEMRI salary scale, in accordance with the structure approved by the Kenyan Salaries and Remuneration Commission (SRC). All costs were converted to USD using the exchange rate from the OANDA currency converter (https://www.oanda.com/currency-converter/en/)

Line 157-161 – Do you have a reference to sustain the 10% undetermined results? Avoid speculation. If that is in-house data, add that to the results and discuss it.

Line 157-161 – A reference [21] has been included.

Lines 179-180 – Random figure 1 legend. It was confusing to see the incomplete legend in the middle of the text.

This has been edited accordingly. The figure legend has been removed.

Line 229 – BM? Please edit table 2 on “blood meal” for consistency.

This has been edited accordingly

Line 245 – Anything else besides Table 2?

Labor cost is now in table 2

Line 259 – Investigation of a single hypothetical scenario, correct? Please remove the plural from the title.

This line has been revised accordingly

Line 239 – Remove bold from Direct on Table 4.

This line has been revised accordingly

Lines 262 and 265 - Table 5 typo.

Corrected accordingly

Line 266 – Correct to: “...the material, labor, total direct costs, and time-to-results...”. Also remove the bold from Table 5.

This line has been revised accordingly

Line 287 – Is it sensitivity or accuracy? Please reword.

This line has been revised accordingly

Line 299 – Bold statement. Due to several limitations regarding processing of complex samples (3-component samples – pathogen-vector-host), the use of MALDI-TOF for any vector-borne has several complicators not part of most medical microbiology systems. Please rephrase accordingly.

This statement has been rephrased accordingly

---

## [Editor Report · Decision Letter 1]

15 Oct 2025

MALDI-TOF MS for malaria vector surveillance: a cost-comparison analysis using a decision-tree approach

PONE-D-25-38538R1

Dear Dr. Karisa,

We’re pleased to inform you that your manuscript has been judged scientifically suitable for publication and will be formally accepted for publication once it meets all outstanding technical requirements.

Kind regards,

Bersissa Kumsa, DVM, MSc, PhD

Academic Editor

PLOS ONE
---

## [Editor Report · Acceptance letter]

PONE-D-25-38538R1

PLOS ONE

Dear Dr. Karisa,

I'm pleased to inform you that your manuscript has been deemed suitable for publication in PLOS ONE. Congratulations! Your manuscript is now being handed over to our production team.

Kind regards,

on behalf of

Professor Bersissa Kumsa

Academic Editor

PLOS ONE